# Characteristics and Biological Activity of Exopolysaccharide Produced by *Lysobacter* sp. MMG2 Isolated from the Roots of *Tagetes patula*

**DOI:** 10.3390/microorganisms10071257

**Published:** 2022-06-21

**Authors:** Inhyup Kim, Geeta Chhetri, Yoonseop So, Jiyoun Kim, Taegun Seo

**Affiliations:** Department of Life Science, Dongguk University-Seoul, Goyang 10326, Korea; inhyup91@gmail.com (I.K.); geetachhetri123@yahoo.com (G.C.); sts5552@naver.com (Y.S.); ginapd@daum.net (J.K.)

**Keywords:** *Lysobacter*, exopolysaccharide, antibiofilm activity, rhizosphere, *Tagetes*

## Abstract

In the present study, exopolysaccharide (EPS) produced by *Lysobacter* sp. MMG2 (lyEPS) was characterized and purified. The lyEPS-producing strain *Lysobacter* sp. MMG2 was isolated from the roots of *Tagetes patula*. When lyEPS was produced in tryptic soy broth with 1% glucose and the lyophilized powder was measured, the yield was found to be 0.67 g/L. The molecular weight (Mw) of lyEPS was 1.01 × 105 Da. Its monosaccharide composition includes 84.24% mannose, 9.73% glucose, 2.55% galactose, 2.77% arabinose, 0.32% xylose, and 0.03% rhamnose. Scanning electron microscopy (SEM) revealed that lyEPS has various round and rough surfaces. Fourier-transform infrared (FTIR) analysis identified its carbohydrate polymer functional groups. Moreover, thermogravimetric analysis of lyEPS revealed two events of mass loss: the first was water loss, which resulted in 3.97% mass loss and the second event occurred at approximately 212 °C. lyEPS could inhibit biofilm-producing pathogenic bacteria without any antimicrobial activity. Furthermore, lyEPS at a concentration of 4 mg/mL could exhibit potent 2,2-diphenyl-1-picrylhydrazyl (DPPH) free radical-scavenging activity (89.25%). These results indicate that lyEPS could be a promising candidate for industrial development if its biological activity is further explored.

## 1. Introduction

Exopolysaccharides (EPSs) are long-chain high-molecular biopolymers that are branched and repeating units of sugars or sugar derivatives [1]. EPS is produced by metabolic processes in microorganisms, including bacteria, blue-green algae, and fungi [2]. EPSs are abundant in the rhizosphere and have a great impact on the industry [3,4]. Endogenous bacteria in plants are important because of their potential to synthesize and produce novel bioactive polymers [5,6]. Bacteria produce biopolymers with different chemical properties. Some of these biopolymers have bioactive functions [7]. For example, biofilms may exhibit beneficial effects against antimicrobial resistance or drought stress by inducing microbial cell adhesion in certain plants [8]. In recent years, bacterial EPSs have attracted attention because of their various biological activities, such as antibacterial, antiviral, immunostimulatory, anti-inflammatory, antitumor activity, and wound healing activities [9,10,11,12,13,14,15].

More than 80% of microbial infections in humans are caused by bacterial biofilms [16]. The Gram-positive bacteria *Staphylococcus epidermidis* and *S. aureus*, which account for more than 50% of species isolated from patients with medical device-related infections, are examples of biofilm-producing strains [17]. Biofilm-forming bacteria are difficult or impossible to eradicate as they are protected from antibiotics or host defenses [18]. We assessed lyEPS produced by *Lysobacter* sp. MMG2 and found that it could inhibit biofilm formation in *S. aureus* subsp. *aureus* ATCC 6538, *S. epidermidis* KACC 13234, and *Pseudomonas aeruginosa* ATCC 15442. As lyEPS extracted and purified from rhizospheres could inhibit biofilm formation in pathogens, it can be considered a good option for inhibiting biofilm adhesion.

Reaction oxygen species (ROS) is a byproduct of normal oxygen metabolism and is a highly reactive chemical known for homeostasis and cell signaling [19]. Negative effects of ROS include cancer and the aging process [20,21]. Even DNA damage alone increases ROS levels within the cell, so reducing these negative aspects with an antioxidant system can improve efficiency for the good [22]. DPPH is a substance with free radicals, and it is widely used in antioxidant analysis [23]. Several studies have been conducted to investigate this antioxidant performance, as it can reduce the aforementioned negative aspects and only highlight the positive aspects [24,25,26]. We found that lyEPS produced by *Lysobacter* sp. MMG2 has free-radical scavenging activity and, upon reaching a particular concentration, has a free-radical scavenging activity similar to that of ascorbic acid.

The genus *Lysobacter* belonging to the family *Lysobacteraceae* within the class *Lysobacterales*, was first proposed by Christensen and Cook as non-fruiting, gliding bacteria [27]. Most strains of the *Lysobacter* genus are Gram-stain-negative, aerobic, non-fruiting, gliding bacteria. *Lysobacter* species have been isolated from various environments such as soil, seawater, sludge, plant, sediment, and estuary sediments [28,29,30,31,32,33]. The isolate *Lysobacter* sp. MMG was excavated from roots of *Tagetes patula* (marigold) in Ilsan, Republic of Korea, and *Tagetes patula* are known as medicinal plants [34]. *Lysobacter* sp. XL1 produces an enzyme complex lysoamidase and is used as a drug to treat Gram-positive bacterial infections [35]. In addition, some *Lysobacter* species are reported to exhibit antibacterial and antifungal activity, and *Lysobacter enzymogenes* exhibit biological control activity [36,37,38,39,40].

In the present study, we identified and characterized lyEPS obtained from rhizospheres. To the best of our knowledge, no previous study has reported the characterization of lyEPS, which is produced by *Lysobacter* isolated from the medicinal plant *Tagetes patula* (marigold). Therefore, in the present study, we assessed the monosaccharide composition, and molecular weight were evaluated through FTIR analysis, bioliquid chromatography (Bio-LC), and gel permeation chromatography (GPC) and functional groups were also identified. In addition, we performed in vitro analysis to assess the antibiofilm and antioxidant potentials of lyEPS. Our findings provide theoretical support for the industrial applications of EPS obtained from rhizospheres.

## 2. Materials and Methods

### 2.1. Sample Collection and Isolation of Strain

Roots of marigold plants (*T. patula*) were harvested from a garden near Dongguk University in Goyang, South Korea (37°40′26.4′′ N, 126°48′20.88′′ E). The roots were washed with 70% ethanol and sterile water to remove external dirt and dust from the surface. For isolation, the root sample (10 g) was crushed using a sterilized mortar and mixed in 5 mL sterile phosphate-buffered saline (PBS), pH 7.4 (10 mM, Welgene, Gyeongsan, South Korea). Then, the solution was serially diluted up to 10^−5^ using a previously described standard method [41,42]. An 800 µL aliquot of this sample was spread onto tryptic soy agar (TSA) plates (Difco) supplemented with 1% glucose (*v*/*w*; Biopure, Cambridge, MA, USA) and incubated at 30 °C for 7 days. Colonies were subsequently selected to purify strains suspected of secreting EPS. The isolates were stored in tryptic soy broth inoculated with 25% glycerol (*v*/*v*; Sigma-Aldrich, St. Louis, MO, USA) at −80 °C.

### 2.2. 16S rRNA Gene Sequencing and Culture Conditions

DNA extraction was performed using the TaKaRa MiniBEST Bacterial Genomic DNA Extraction Kit ver. 3.0 (Takara Bio, Kusatsu, Japan), according to the manufacturer’s protocol. The 16S rRNA gene sequences of the isolates were amplified using the universal bacterial primer sets 27F and 805R as well as 518F and 1492R, and sequencing was then performed [43]. The 16S rRNA gene sequences were assembled using SeqMan 2 software (DNASTAR). Multiple sequences were aligned using MEGA X software (Version: 11) program and analyzed using CLUSTAL X [44,45]. Phylogenetic trees were constructed according to the neighbor-joining (NJ) and maximum-likelihood (ML) methods using the Kimura two-parameter model [46]. The min-mini heuristic algorithm was applied in the maximum-parsimony (MP) method to compare the phylogenetic trees constructed using the neighbor-joining method [47]. The phylogenetic tree was assessed by bootstrap analysis with 1000 replications [48].

The EzBioCloud.net, available online: http://www.ezbiocloud.net (accessed on 30 May 2022) and NCBI BLAST databases were used to assess the 16S rRNA gene sequencing results [49,50]. The growth of strain MMG2 on Reasoner’s 2A agar (R2A agar; Difco), Luria–Bertani agar (LB agar; Difco), nutrient agar (NA; Difco), and TSA was assessed at 10, 15, 20, 25, 30, 37, and 42 °C for 14 days to determine the optimal culture condition. All these media were supplemented with 1% glucose (*v*/*w*).

### 2.3. Extraction and Purification of lyEPS

To extract EPS, 1 L of tryptic soy broth supplemented with 1% glucose was autoclaved in a 2 L Erlenmeyer flask. Strain MMG2 of the culture of 1% (*v*/*v*) EPS-producing was strained overnight (optical density (OD) at 600 nm = 0.5) and incubated for 7 days at 30 °C on a shaking incubator (160 rpm) for EPS production. The culture was centrifuged at 6000× *g* for 30 min at room temperature to remove the biomass. Following this, 14% trichloroacetic acid (Duksan Pure Chemicals, Ansan-si, South Korea) was added to the supernatant and incubated for 30 min at 90 rpm. The mixture was then centrifuged at 6000× *g* for 20 min to remove the denatured protein. Ice-cold absolute ethanol (Merck, stored at −18 °C) was added at twice the volume of the supernatant and precipitated overnight in a refrigerator at 4 °C. The obtained EPS solution was dialyzed with ultrapure water (UPW) using a dialysis tubing bag (10K MWCO, SnakeSkin Dialysis Tubing). The purified EPS solution was lyophilized to obtain a powder, and the weight was recorded.

### 2.4. Determination of Monosaccharide Composition and Chemical Analysis of lyEPS

After hydrolysis of 2 mg of the EPS sample with 2 mL of 2 M trifluoroacetic acid, the monosaccharide composition was analyzed by Bio-LC on a Dionex™ Carbopac™ PA-20 anion exchange chromatography column (ICS-5000PC, Thermo Dionex, Germany). For peak identification, standards of mannose, glucose, galactose, arabinose, and rhamnose were used.

The total sugar content of lyEPS was determined by the phenol sulfate method using glucose as the standard according to the previously described method [51]. The protein content of lyEPS was determined using the Lowry assay, and bovine serum albumin (BSA) was used as a calibration standard [52]. The standard curves were created according to the manufacturer’s instructions with various concentrations of BSA (2 mg/mL, 1.5 mg/mL, 1 mg/mL, 0.75 mg/mL, 0.5 mg/mL, 0.25 mg/mL, 0.125 mg/mL, 0.025 mg/mL, and 0 mg/mL) (Pierce^TM^ BCA Protein Assay Kit; Thermo Fisher Scientific, Waltham, MA, USA). All values were measured in absorbance at 562 nm by spectrophotometer (Multiskan GO; Thermo Fisher Scientific, Waltham, MA, USA).

### 2.5. SEM Analysis and Determination of Mw of lyEPS

The surface morphology and microstructure of the lyophilized lyEPS (3 mg) were visualized using a field-mission scanning electron microscope (Sigma; Carl Zeiss, UK) at an accelerating voltage of 15 kV. lyEPS was mounted on a stub coated with a gold coating unit (10 nm; EM ACE200, Leica).

The Mw of lyEPS was assessed by gel permeation chromatography (GPC) (HLC-8420; Tosoh, Japan) with a TSKgel G2500PW_XL_ column and estimated using an RI detector. lyEPS solution (3 mg/mL; 50 μL) was injected and eluted with 0.1 M NaNO3 at 40 °C at a flow rate of 1 mL/min [53]. The Mw of the sample was calculated using EcoSEC Elite-WS software based on the peak time.

### 2.6. FTIR Analysis of lyEPS

FTIR–attenuated total reflection (ATR) spectra of the lyEPS samples were assessed to identify characteristic functional groups. Spectra were obtained from 4000 cm^−1^ to 400 cm^−1^ on a Perkin Elmer spectrophotometer using the diamond crystal ATR method (background scans, 32; resolution, 3).

### 2.7. Thermogravimetric Analysis (TGA) and X-ray Diffraction (XRD) Analysis of lyEPS

To assess the physical properties of lyEPS, XRD analysis and TGA were performed using the lyophilized lyEPS powder. TGA of lyEPS was performed using Pyris TGA N-1000. For this purpose, 10 mg of the lyEPS sample was heated from 25 °C to 800 °C under the flow of nitrogen air at 10 °C/min.

XRD analysis was performed at a range of 5 °C to 80 °C (Ultima; Rigaku, Tokyo, Japan). The lyEPS sample was pulverized into a fine powder and mounted on a quartz substrate. Cu kα X-rays were then generated to record intensity peaks continuously using a scintillation counter detector.

### 2.8. Determination of Antibiofilm Activity of lyEPS

The inhibition of biofilm formation by lyEPS was assessed using the method as described previously, with slight modifications [54,55,56]. In brief, 1.0 × 10^−8^ CFU/mL of *S. aureus* subsp. *aureus* ATCC 6538, *S. epidermidis* KACC 13234, and *P. aeruginosa* ATCC 15442 cultures were added alone to 96-well plate at 100 μL. Following this, 100 μL of lyEPS at 1, 2, 4, 6, 8, and 10 mg/mL concentrations were added to wells in a 96-well microtiter plate. The sterile culture medium was used as a negative control. After 24 h of incubation at 37 °C, nonadherent bacteria were washed thrice with distilled water and dried. Staining was performed for 10 min by adjusting the concentration of crystal violet to 2%, followed by washing thrice in distilled water to remove the dye. Then, 160 µL of 33% acetic acid was added to each well in order to resolubilize the dye and transferred to a new 96-well plate. Using a spectrophotometer (Multiskan GO; Thermo Fisher Scientific, Waltham, MA, USA), the OD of the biofilm mass was measured at 590 nm.

### 2.9. Determination of Antioxidant Activity of lyEPS

The antioxidant activity of lyEPS was assessed using the DPPH free radical-scavenging assay according to a previous report, with slight modifications [57]. In brief, 1.25 mL of DPPH solution (0.2 mM) was mixed with 0.25 mL of lyEPS sample solutions at various concentrations. After incubation at room temperature for 30 min, 200 μL was transferred to a 96-well microplate and assessed using a microplate reader in the 517 nm range (absorbance). The antioxidant activity of lyEPS was measured in triplicate. Ascorbic acid was used as a positive control. The DPPH free radical-scavenging activity was calculated according to the following equation:Scavenging activity (%) = [1 − (A_sample_ − A_blank_)/A_control_] × 100%(1)
where A_sample_ is the absorbance of DPPH solution mixed with the sample, A_blank_ is the absorbance of DPPH solution, and A_control_ is the absorbance of the control.

## 3. Results and Discussion

### 3.1. Culture Conditions and 16S rRNA Gene Sequencing

The ability of the selected strains to produce EPS was primarily screened by subculture on TSA plates supplemented with 1% glucose (*v*/*w*). Moreover, EPS production was assessed using the phenol sulfate method [51]. Among the selected strains, strain MMG2 was selected for the present study as it produced more EPS. Among the four media, the maximum amount of lyEPS was detected on TSA plates supplemented with 1% glucose. Hence, tryptic soy broth supplemented with 1% glucose was used for the mass production of lyEPS. Moreover, 16S rRNA gene sequencing was performed. NCBI BLAST and EzBioCloud database search results revealed that strain MMG2 belonged to the genus *Lysobacter* and was closely related to *Lysobacter parti* SYSU H10001 29487 (99.25%). The type species is *Lysobacter enzymogenes* ATCC 29487 (Figure 1).

### 3.2. lyEPS Production

The growth of strain MMG2 on TSA plates occurred at 10–42 °C. The strain grew on R2A agar, NA, and LB agar; however, the growth was slower than that on TSA. *Lysobacter* sp. MMG2 produced EPSs in all the media tested in the present study. lyEPS was extracted by culturing the strain at 30 °C for 7 days. Extensive cell growth was observed after 24 h of inoculation. However, after 168 h of inoculation, the cells reached a stationary phase. After culturing the strain in tryptic soy broth with 1% glucose for 168 h, lyEPS was extracted and purified. After inoculation with *Lysobacter* sp., MMG2 (OD590 = 0.5) in fresh culture medium was incubated at 30 °C for 240 h with shaking. At 24 h intervals, the absorbance of 200 μL of cultures was measured at OD590. When the absorbance no longer increases, it is judged as a stationary period. The lyEPS was weighed after lyophilization and the yield was 0.67 g/L.

### 3.3. Monosaccharide Composition of lyEPS

The monosaccharide composition of lyEPS was assessed using a Bio-LC system (ICS-5000PC; Thermo Dionex, Germany). As shown in Figure 2, acid-hydrolyzed lyEPS contained the highest amount of mannose (84.25%), followed by glucose (9.73%), galactose (2.88%), arabinose (2.77%), xylose (0.32%), and rhamnose (0.03%). It is known that the monosaccharide composition of EPS is affected by strain growth conditions and medium composition [58]. Changes in the biochemical properties and variability in the monosaccharide composition of EPS may contribute to bacterial adhesion and proliferation [59]. As xylose and arabinose are not normally present in bacterial EPS, lyEPS exhibits several properties that differ from normal EPS [60,61]. EPS containing rhamnose is known to have superior biological properties to EPS lacking rhamnose, which is known for its general monosaccharide composition. lyEPS contains a small amount of rhamnose, a rare sugar [62]. Results similar to the monosaccharide composition of lyEPS have been reported in *Lactobacillus plantarum* [63]. EPSs containing xylose and arabinose were also present in EPSs generated from the two strains *Rhodothermus marinus* DSM 4252 and *Rhodothermus marinus* MAT 493 studied by Sardari et al. [64]. However, the monosaccharide composition of lyEPS and EPS of the two strains are different, indicating different usefulness of EPS from the three strains. The percentage of carbohydrate and protein content of lyEPS fractions was 72.8% (*w*/*w*) and 6.2% (*w*/*w*), respectively.

### 3.4. SEM Analysis and Mw of lyEPS

SEM helps to recognize the surface and microstructure, representing the three-dimensional structure of lyEPS. It is an exceptional tool for understanding the physical properties of macromolecules [65]. The surface morphology of lyEPS was dense and irregular. It was found to be a relatively stable three-dimensional structure, showing a rough surface with irregular lumps of various sizes at various magnifications (Figure 3a–d).

The Mw of polysaccharides is an important parameter that influences their functional properties [66]. To calculate the Mw of lyEPS, the molecular weight was calculated from a calibration curve using various pullulan (Mw: 180 Da, 667 Da, 6.3 kDa, 9.8 kDa, 22 kDa, 49.4 kDa, 110 kDa, 201 kDa, 334 kDa, and 642 kDa) as standard material, as described previously [1]. The GPC was performed by injecting a solution of lyEPS (3 mg/mL; 50 μL) eluted with 0.1 M NaNO3 at 40 °C at a flow rate of 1 mL/min [53]. The Mw of lyEPS was estimated to be 1.01 × 10^5^ Da, and the GPC results of lyEPS revealed only one fraction (Figure 3e). The Mw of the isolated EPS could depend on various factors, such as the nature of the starting material, extraction temperature, and fractionation method employed [67].

### 3.5. FTIR Analysis of lyEPS

The FTIR–ATR spectra of lyEPS revealed various functional groups (Figure 4). In lyEPS, the broad absorption band at 3284.81 cm^−1^ represents the stretching vibration of O-H and that at 2924.84 cm^−1^ represents the bending and stretching vibrations of C-H [68,69]. Polysaccharides contain numerous hydroxyl groups and exhibit strong and extensive stretching vibration in that region [70,71]. The adsorption at 1022.55 cm^−1^ can be attributed to Si-O or C-O bonds that are known to be associated with polysaccharides, alcohols, and ethers. Peaks ranging from 1022.55 cm^−1^ to 2924.84 cm^−1^ may also represent silica [72]. Moreover, the strong absorption band at 1022.55 cm^−1^ is indicative of a polysaccharide [70]. Absorption in the region at 1636.99 cm^−1^ is associated with the C = O group [73].

### 3.6. TGA and XRD Analysis of lyEPS

The first mass loss event occurred between 27 °C and 68 °C, with the maximum mass loss (3.97%) occurring at 47.4 °C (Figure 5). This mass loss event could be related to water loss [74]. Above 68 °C, EPS energy began to be released. The second event occurred at approximately 212 °C. Energy release occurred, with the maximum exothermic peak being noted at 246.1 °C and representing 75.9% loss of the initial mass. The temperature at which the mass loss rate of lyEPS became 50% was 356.7 °C. Thermostability is an important property of EPS that can have industrial and commercial applications. For example, to determine whether EPSs can be commercially used in the food industry or can be used as coatings to deliver specific drugs, it is necessary to determine whether they are thermostable [75,76]. Although the initiation temperature of the initial thermal reaction of lyEPS was low, the temperature at which the thermal mass loss reached 50% was as high as 356.7 °C; this temperature is higher than that required for previously reported bacterial EPSs [77]. Thus, the thermostability of lyEPS makes it a promising candidate for use in the food industry. In the present study, TGA of EPSs produced by the genus *Lysobacter* was performed for the first time.

XRD is commonly used for the phase identification of materials. XRD can be used for the qualitative and semiquantitative evaluations of amorphous and crystalline compounds. XRD analysis of the lyophilized lyEPS powder was performed to determine its crystalline/amorphous properties. A typical amorphous pattern was detected (Figure 6). The XRD pattern had a broad peak around approximately 10–30 °C (2θ), indicating the presence of amorphous properties [78]. The XRD of EPS studied by Rani et al. also showed an amorphous pattern [79]. Beyond that, previous studies showed amorphous patterns, as in Singh et al. [80] and Bhat et al. [81].

### 3.7. Antibiofilm Activity of lyEPS

Different concentrations (0.25, 0.5, 1, 2, and 4 mg/mL) of lyEPS showed no antibacterial effect against *S. aureus* subsp. *aureus* ATCC 6538, *S. epidermidis* KACC 13234, and *P. aeruginosa* ATCC 15442. Thus, lyEPS exhibited antibiofilm activity without any antibacterial activity. The antibiofilm activity of lyEPS was concentration dependent. The higher the absorbance of the bacterial suspension, the greater was the quantity of the strain. Absorbance measurement facilitated the qualitative analysis of the antibacterial film of lyEPS. The antibiofilm effect of lyEPS on *S. aureus* subsp. *aureus* ATCC 6538, *S. epidermidis* KACC 13234, and *P. aeruginosa* ATCC 15442 was quantitatively assessed by measuring the absorbance of each of the three strains. When lyEPS was applied to *S. epidermidis* KACC 13234 at a concentration of 4 mg/mL, the biofilm inhibition rate was the highest at 96.0% (Figure 7). The results of Jiang et al.’s study were similar to our lyEPS in that A101 EPS generated from *Vibrio* sp. QY101 showed antibiofilm activity without antimicrobial activity [54].

*S. aureus* biofilms are known to be associated with intramammary infections [82]. The antibiofilm activity of lyEPS may make it a candidate for reducing antimicrobial susceptibility to biofilms of pathogenic strains, such as *S. aureus*, *S. epidermidis*, and *P. aeruginosa*, and preventing damage from medical device attachment [83].

### 3.8. DPPH Free Radical-Scavenging Activity of lyEPS

Natural antioxidants that maintain human health and prevent or treat diseases have attracted considerable attention [84]. Polysaccharides have been considered promising antioxidants as candidates for developing nontoxic pharmaceuticals with strong antioxidant effects in vitro and in vivo [85].

DPPH free radicals are widely used as tools for assessing the free radical-scavenging activity. The DPPH free radical-scavenging activity of lyEPS was assessed at various concentrations (0, 0.25, 0.5, 1, 2, and 4 mg/mL). Ascorbic acid at the same concentrations was used as a positive control (Figure 8). At a concentration of 0.25 mg/mL, lyEPS and ascorbic acid exhibited 76.88% and 95.36% DPPH free radical-scavenging activity, respectively, with the scavenging activity of lyEPS being lower. However, as the lyEPS concentration increased to 4 mg/mL, its DPPH free radical-scavenging activity increased to 89.25%, reaching significantly closer to that of ascorbic acid at the same concentration (96.5%). The DPPH free radical-scavenging activity of lyEPS increased with an increase in its concentration, becoming similar to that of ascorbic acid (a difference of less than 7.3%).

To the best of our knowledge, this is the first study of the DPPH radical scavenging activity of EPS produced by *Lysobacter* species. So we compared our results with EPSs generated by different genera. While the DPPH scavenging activity of lyEPS at a concentration of 4 mg/mL was 89.25%, at a concentration of 10 mg/mL of SSC-12 EPS produced from *Pediococcus pentosaceus* SSC-12, the DPPH scavenging ability of SSC-12 EPS was 77.4%, which was lower than that of lyEPS DPPH scavenging activity [86]. The DPPH scavenging activity of lyEPS 4 mg/mL (89.25%) was higher than that of EPS produced by *Pseudomonas fluorescens* CrN6 (concentration, 4 mg/mL; DPPH scavenging activity, 45.4%) isolated from rhizosphere [87]. On the other hand, a level similar to the DPPH scavenging activity of EPS produced by *Bacillus megaterium* PFY-147 was observed [88].

## 4. Conclusions

lyEPS, obtained from *Lysobacter* sp. MMG2 isolated from the roots of *T. patula*, contains mannose, glucose, galactose, arabinose, xylose, and rhamnose. Purified lyEPS was characterized by GPC, FTIR analysis, SEM, Bio-LC, TGA, and XRD analysis in the present study. lyEPS inhibited biofilm formation, exhibited remarkably high antioxidant activity, and had excellent thermostability. Therefore, it has the potential to be used in several industries. The present study is the first to report the characterization of EPSs obtained from the genus *Lysobacter*.

## Figures and Tables

**Figure 1 microorganisms-10-01257-f001:**
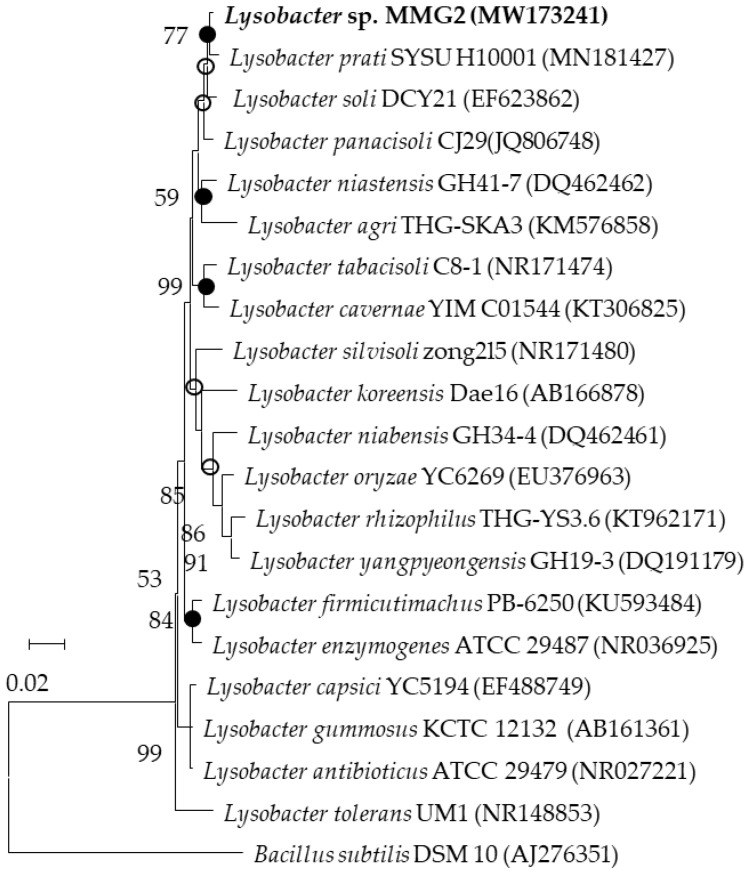
Phylogenetic tree of strain MMG2 within the genus *Lysobacter*. Bootstrap values are shown as percentages of 1000 replicates (only values > 50% are shown). Filled circles indicate the corresponding nodes were recovered in trees generated with the ML and MP algorithm. Empty circles indicate the corresponding nodes were recovered using the ML algorithm. Bar, 0.02 substitutions per nucleotide position.

**Figure 2 microorganisms-10-01257-f002:**
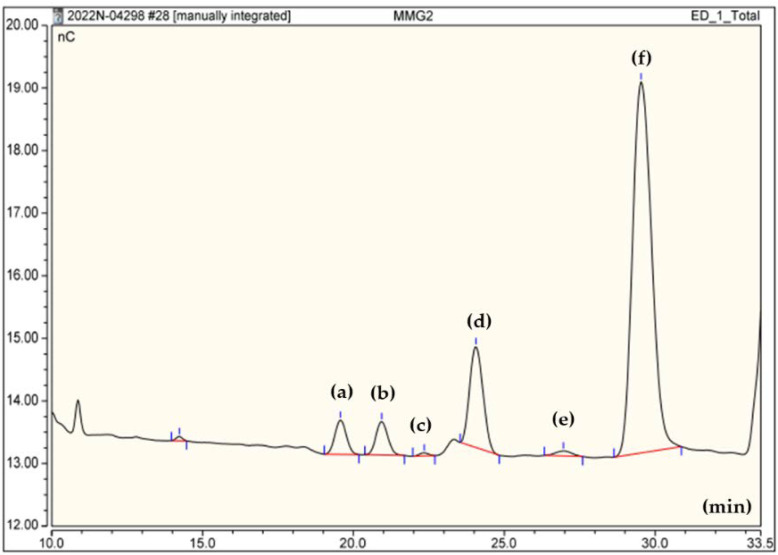
Assessment of monosaccharide composition of EPS produced by *Lysobacter* sp. MMG2 (lyEPS) by Bio-LC (Dionex ICS-5000PC. (**a**) Arabinose (19.592), (**b**) galactose (20.942), (**c**) rhamnose (22.292), (**d**) glucose (24.067), (**e**) xylose (26.926), and (**f**) mannose (29.567). The retention time for absorption peaks is indicated in parentheses in minutes. The monosaccharide composition of lyEPS was in the following descending order: mannose, glucose, galactose, arabinose, xylose, and rhamnose.

**Figure 3 microorganisms-10-01257-f003:**
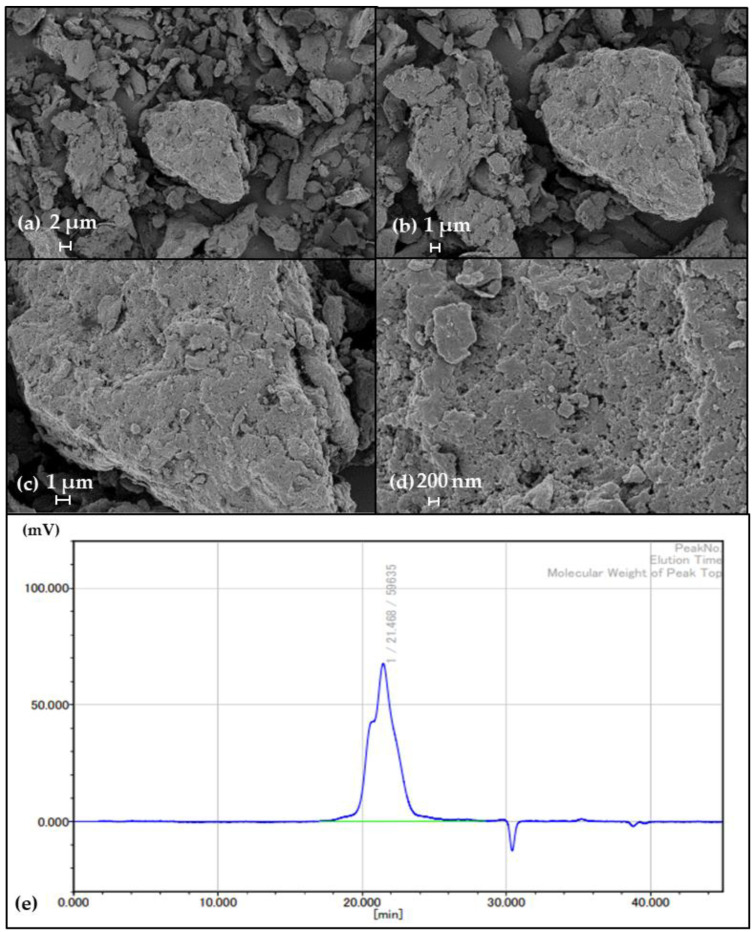
SEM images of purified lyEPS. The magnifications of the SEM images are (**a**) 3000×, (**b**) 5000×, (**c**) 10,000×, and (**d**) 30,000×; (**e**) Mw distributions of lyEPS.

**Figure 4 microorganisms-10-01257-f004:**
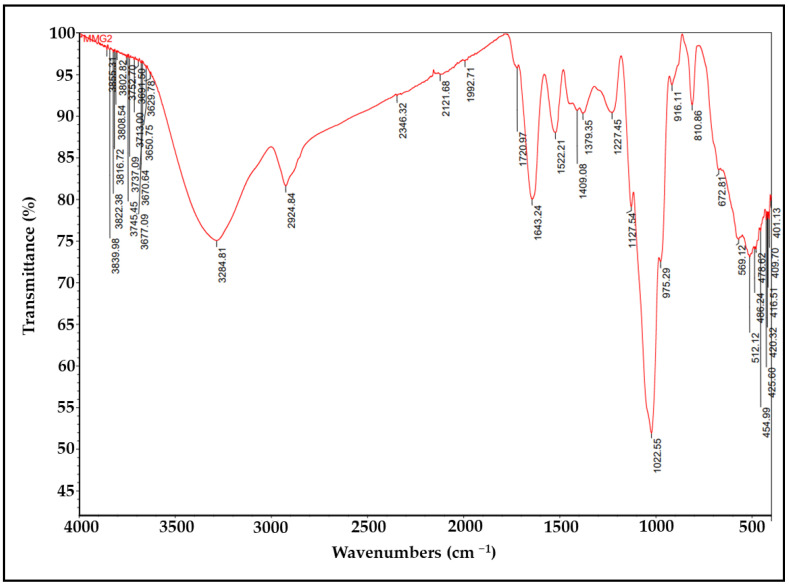
FTIR analysis of EPS produced by *Lysobacter* sp. MMG2 (lyEPS).

**Figure 5 microorganisms-10-01257-f005:**
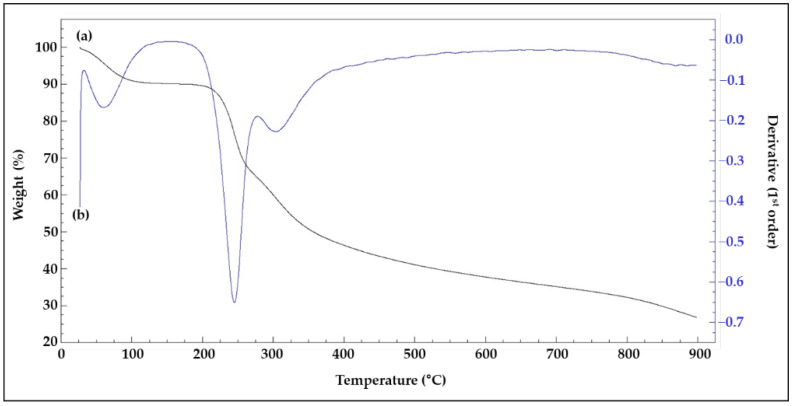
TGA analysis of purified lyEPS produced after 7 days of incubation of *Lysobacter* sp. MMG2 in tryptic soy broth supplemented with 1% glucose. (**a**) TGA and (**b**) DTA of lyEPS.

**Figure 6 microorganisms-10-01257-f006:**
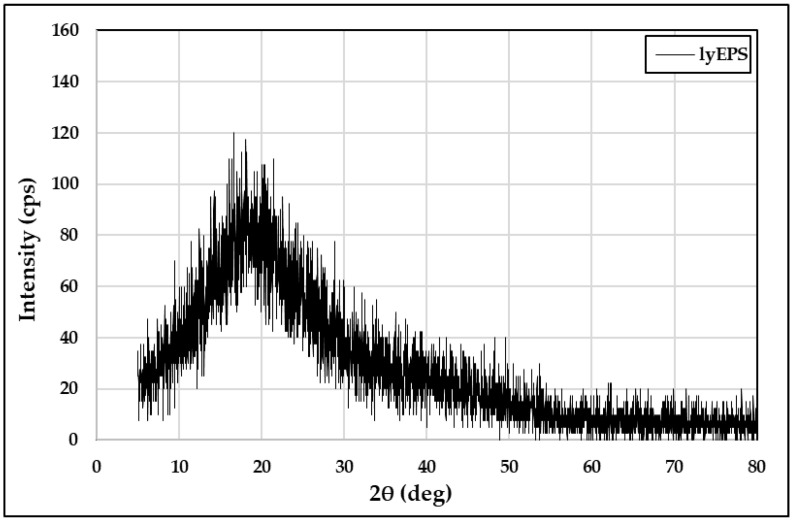
XRD spectra of EPS produced by *Lysobacter* sp. MMG2 (lyEPS).

**Figure 7 microorganisms-10-01257-f007:**
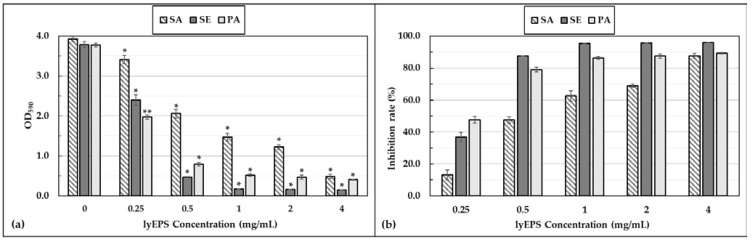
Antibiofilm activity of lyEPS at different concentrations (0, 0.25, 0.5, 1, 2, and 4 mg/mL). The biofilm inhibitory effect of lyEPS on *Staphylococcus aureus* subsp. *aureus* ATCC 6538, *Staphylococcus epidermidis* KACC 13234, and *Pseudomonas aeruginosa* ATCC 15442 was assessed at OD590. It shows that the OD590 measured value decreases with the increasing concentration of lyEPS (**a**). The inhibition rate was calculated as the relative increase (%) in comparison with the control group (0%, without lyEPS). The inhibition rate increased with increasing lyEPS concentration (**b**). Significant differences between treatments and the control were represented by “*”, *p* < 0.01; “**”, *p* < 0.001.

**Figure 8 microorganisms-10-01257-f008:**
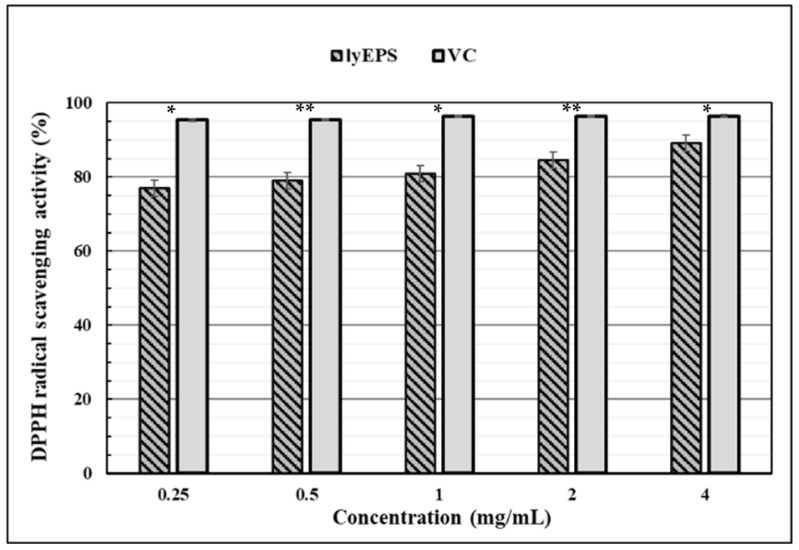
DPPH free radical-scavenging activity of EPS produced by *Lysobacter* sp. MMG2 (lyEPS). All experiments were conducted in triplicate, and data are presented as mean ± SD. The significance of lyEPS and ascorbic acid (VC) values at the same concentration was represented by “*****”, *p* < 0.01; “******”, *p* < 0.001.

## Data Availability

The data that support the findings of this study are available from the corresponding author upon reasonable request.

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
