# Peer review of "Characteristics and Biological Activity of Exopolysaccharide Produced by Lysobacter sp. MMG2 Isolated from the Roots of Tagetes patula"

_microorganisms, 2022, doi:10.3390/microorganisms10071257_

Round 1

Reviewer 1 Report

I have reviewed the Manuscript titled: ‘Characteristics and biological activity of exopolysaccharide produced by Lysobacter sp. MMG2 isolated from the roots of Tagetes patula’ and have the following comments.

In Section 2.8 please explain how the bacterial cultures were added to the 96-well plate- singly or in combination an the concentration added. Also mention the controls used.

In Section 3.1 the sequence comparison should only be done against ‘type’ strains and not the others as type strains have been through a more vigorous process to be listed as a ‘Valid Name”

In Section 3.2 please explain how growth was measured and how was the stationary phase determined.

In Section 3.4 please give details of the chromatography such as eluting solvents, flow rates, and whether standards were used to calibrate the concentration and Rt of the sugars.

In Section 3.5 please explain what is meant by ‘represent CO2 from respiration when FTIR was performed.”

In Sections 3.7 and 3.8 was statistical analysis applied to determine whether there were significant differences  between the treatments.

Reviewer 2 Report

This paper titled “Characteristics and biological activity of exopolysaccharide produced by Lysobacter sp. MMG2 isolated from the roots of Tagetes patula” is interesting. But, more experiments are requested. Thus, I think this manuscript could be considered for publication in Microorganisms after Major revising. 

My comments are as follow:

  1. Check the abbreviations, the text format of full text. 
  2. The introduction section is too simple.
  3. The chemical analysis of the exopolysaccharide was requested. The monosaccharides composition and properties of exopolysaccharide should be explored.
  4. Please compared your results to previous studies. 

Round 2

Reviewer 1 Report

The response to my first question is poor: Change: Following this, 100-μL of lyEPS at 1, 2, 4, 6, 8, and 10 mg/mL concentrations were added to wells in a 96-well microtiter plate.

The sterile culture medium was used as a negative control.

Line  200: The authors claim that strain MMG2 belongs to the genus Lysobacter and was closely related to Lysobacter enzymogenes ATCC 29487 (96.9%) when in fact it is closest to Lysobacter prati SYSU H10001 (99.25%) as show in the phylogenetic tree. Details of the tree construction (algorithm used) should be provided. In fact, best practice is to run at least two algorithms to determine relationships with other known valid species.

In your response to my comment on monitoring growth it appears that this was done in a fresh medium with no inoculum -lines 216-218.

Line 261 the peak is not symmetrical in Fig 3 e. Also size bars are required for the micrographs in Figs 3 a-d.

With regards to stats analysis just adding a sentence at the end of the caption of Fig 7 -"All experiments were conducted in triplicate, and data are presented as mean ± SD. (p < 0.01)." has no meaning.

In Fig 8 what does the symbol "VC" stand for?

Reviewer 2 Report

The authors addressed all my comments.

Author Response

Thank you for your valuable comments.